# A New Insight into Reliability Data Modeling with an Exponentiated Composite Exponential-Pareto Model

Bowen Liu [†,‡] and Malwane M. A. Ananda *,[†,‡]

Department of Mathematical Sciences, University of Nevada, Las Vegas, NV 89154, USA
* Correspondence: malwane.ananda@unlv.edu
† Current address: 4505 S Maryland Pkwy, Las Vegas, NV 89154, USA.
‡ These authors contributed equally to this work.

**Abstract:** It is observed that, for some of the data in engineering and medical fields, the hazard rates increase to a high peak at the beginning and quickly decrease to a low level. In the context of survival analysis, such a hazard rate is called a upside-down bathtub hazard rate. In this paper, we investigated the properties of a model named exponentiated exponential-Pareto distribution. The model was recently proposed and applied to insurance data. We demonstrated that the model has upside-down bathtub-shaped hazard rates with specific choices of parameters. The theoretical properties such as moments, survival functions, and hazard functions were derived. The parameter estimation procedures were also introduced. We then briefly discussed the goodness-of-fit tests of the model with the simulations. Finally, we applied the model to a specific time-to-event data set along with a comparison of the performances with previous existing models. When compared to previous proposed models, the exponentiated exponential-Pareto model demonstrated good performance when fitting to such data sets.

**Keywords:** reliability data modeling; generalized exponentiated composite distributions; exponentiated exponential-Pareto model; upside-down bathtub hazard rate; goodness-of-fit





## 1. Introduction

Reliability data modeling has been an important topic in many different fields such as engineering, biology, and medical sciences. For data sets collected in real life, the hazard rates can show different shapes: monotone increasing, monotone decreasing, bathtub-shaped and upside-down bathtub (UBT)-shaped. While the data sets with monotone or bathtub hazard rates were deeply investigated with different types of parametric models, the data sets with UBT shapes were not as well explored as the others.

In fact, data with UBT-shaped hazard rates appeared frequently in the existing literature. For instance, for the Veteran's Administration lung cancer trial data [1–3], the mortality rate was very low at the beginning, quickly rose to a high peak, and then slowly decreased. This suggests that the data are associated with a UBT-shaped hazard rate. This was also observed in several other data sets from medical and engineering fields [4–6].

To model data with UBT-shaped hazard rates, many parametric models have been proposed in the past. For example, the transmuted Rayleigh distribution [6] was proposed and demonstrated good performance when fitted to a reliability data set with UBT-shaped hazard rates. For more models proposed for this type of data, readers are referred to [7–12].

The concept of the generalized composite distributions was recently proposed to provide better options for fitting right skewed data with a high peak. It was demonstrated that special members from this family, such as exponentiated exponential-Pareto (EEP) distribution and exponentiated inverse-gamma Pareto (EIGP) distribution, can provide satisfactory performances to multiple insurance data sets [13,14]. Moreover, these data sets have very similar features to the ones with UBT-shaped hazard rates. Thus, utilizing these distributions to fit the data sets with UBT-shaped hazard rates seems to be appropriate.

The rest of the paper is organized as follows. Section 2 introduces the concept of the composite distribution. Section 2.2 provides the concept and some properties of the generalized exponentiated composite distribution (GEC). In Section 3, a special model named exponentiated exponential-Pareto (EEP) model is discussed. The parameter estimation of the EEP distribution is introduced in Section 4. In Section 5, by utilizing the goodness-of-fit (GoF) test, a limited simulation study is given to demonstrate the performance of the model associated with the exponent parameter. A real data analysis is presented in Section 6. Finally, the concluding remark and future directions are provided in Section 7.

## 2. Related Work

We start by introducing the concepts of composite distributions and generalized exponentiated composite distributions.

### 2.1. The Composite Distributions

Let $Y$ be a random variable that only takes non-negative real numbers. Let $f_Y(y)$ be the probability density function (pdf) of $Y$. The formal definition of a composite pdf $f_Y(y)$ is given as follows [15]:

$$f_Y(y; \alpha_1, \alpha_2, \theta) = \begin{cases} c f_1(y; \alpha_1, \theta) & 0 \leq y < \theta \\ c f_2(y; \alpha_2, \theta) & y \geq \theta, \end{cases} \tag{1}$$

where $c$ stands for the normalizing constant, $\theta$ is the parameter that denotes the location of probability density change, $f_1$ is the pdf of $Y$ when $Y$ is between 0 and $\theta$, and $\alpha_1$ represents the parameters of $f_1$; $f_2$ is the pdf of $Y$ when $Y$ is greater than $\theta$, and $\alpha_2$ represents the parameters of $f_2$. It is assumed that both $f_1$ and $f_2$ are smooth functions on their supports. $f_1$ is generally called the head pdf of $f_Y$, while $f_2$ is referred to as the tail pdf of $f_Y$.

In real practice, $f_Y(y|\alpha_1, \alpha_2, \theta)$ is usually assumed to be continuous and differentiable. Thus, the following conditions are imposed:

$$\begin{cases} \lim_{y \to \theta^-} f_Y(y; \alpha_1, \alpha_2, \theta) = \lim_{y \to \theta^+} f_Y(y; \alpha_1, \alpha_2, \theta) \\ \lim_{y \to \theta^-} \frac{df_Y(y; \alpha_1, \alpha_2, \theta)}{dy} = \lim_{y \to \theta^+} \frac{df_Y(y; \alpha_1, \alpha_2, \theta)}{dy}. \end{cases}$$

In real practice, the Pareto distribution and the generalized Pareto distribution (GPD) are commonly used for modeling highly right-skewed data at the right tail due the theoretical properties of such distributions. However, both of these distributions can only fit the data beyond a specific threshold. Hence, they cannot be used to globally fit the data.

Many composite models with Pareto or GPD tails have been developed by utilizing the idea of composite distributions, including lognormal-GPD [15], lognormal-Pareto [16], exponential-Pareto [17,18], Weibull–Pareto [19–21], inverse gamma-Pareto [22], and so on. Moreover, Grün and Miljkovic [23] explored a large number of composite distributions with Pareto and GPD tails with a general framework of computational tools.

Composite models with a Pareto tail have been widely used in different fields such as actuarial data and reliability data modeling. While these models were frequently used in the literature, it was found that some of the models such as exponential-Pareto [24] and inverse gamma-Pareto [22] do not perform well when fitting real sets with right skewed distributions. Therefore, a new concept named generalized family of exponentiated composite distribution (GEC) was proposed to address the issue [14].

In the next subsection, we introduce the concept of generalized family of exponentiated composite distributions and the properties of this family.

### 2.2. A Generalized Family of Exponentiated Composite Distributions

A power transformation $T = Y^{\frac{1}{\eta}}$ ($\eta > 0$) on the original composite random variable $Y$ generates an exponentiated composite random variable $T$ with an exponentiated composite pdf. Correspondingly, $T$ is denoted as the exponentiated composite random variable induced by the parent composite random variable $Y$ [14]. The pdf of $T$ can be expressed as follows:

$$f_T(t; \alpha_1, \alpha_2, \theta, \eta) = \begin{cases} cf_1(t^{\eta}; \alpha_1, \theta)\eta t^{\eta-1} & 0 \le t < \theta^{\frac{1}{\eta}} \\ cf_2(t^{\eta}; \alpha_2, \theta)\eta t^{\eta-1} & t \ge \theta^{\frac{1}{\eta}}. \end{cases} \tag{2}$$

The pdf $f_T(t)$ is then referred as the exponentiated composite pdf induced by the parent composite pdf $f_Y(y)$.

Subsequently, we derive the properties of the exponentiated composite random variable $T$ induced by the parent composite random variable $Y$.

#### 2.2.1. Moments

Suppose that a composite random variable $Y$ has finite $k$-th moment $\mu_k$. The $k$-th moment of an exponentiated composite random variable $T$ induced by $Y$ was derived previously by Liu and Ananda [14] as follows:

$$E(T^k) = E(Y^{\frac{k}{\eta}}) = \mu_k^{\frac{1}{\eta}} \tag{3}$$

#### 2.2.2. Survival Function

Assume that $Y$ is a composite random variable with the CDF $F_Y(y)$, where $F_Y(y) = P(Y \le y)$. The CDF of the corresponding exponentiated composite random variable $T = Y^{\frac{1}{\eta}}$ can be represented:

$$\begin{aligned} F_T(t) &= P(T \le t) \\ &= P(Y^{\frac{1}{\eta}} \le t) \\ &= P(Y \le t^{\eta}) \\ &= F_Y(t^{\eta}). \end{aligned} \tag{4}$$

The survival function of $Y$ can then be represented as follows:

$$S_T(t) = 1 - F_T(t) = 1 - F_Y(t^{\eta}). \tag{5}$$

Now, consider the pdf defined in Equation (1). Let $F_1(y; \alpha_1, \theta)$ and $F_2(y; \alpha_2, \theta)$ be the corresponding CDF of $f_1(y; \alpha_1, \theta)$ and $f_2(y; \alpha_2, \theta)$. The CDF of a composite random variable $Y$ can be expressed as follows:

$$\begin{aligned} F_Y(y; \alpha_1, \alpha_2, \theta) &= \int_0^y f_Y(t; \alpha_1, \alpha_2, \theta)dt \\ &= \begin{cases} c\int_0^y f_1(t; \alpha_1, \theta)dt & \text{if } y \in [0, \theta) \\ c[\int_0^{\theta} f_1(t; \alpha_1, \theta)dt + \int_{\theta}^y f_2(t; \alpha_2, \theta)dt] & \text{if } y \in [\theta, \infty) \end{cases} \\ &= \begin{cases} cF_1(y; \alpha_1, \theta) & \text{if } y \in [0, \theta) \\ c[F_1(\theta; \alpha_1, \theta) + F_2(y; \alpha_2, \theta) - F_2(\theta; \alpha_1, \theta)] & \text{if } y \in [\theta, \infty). \end{cases} \end{aligned} \tag{6}$$

Correspondingly, the exponentiated composite random variable $T$ induced by $Y$ has a CDF as follows:

$$F_T(t; \alpha_1, \alpha_2, \theta) = \begin{cases} cF_1(t^\eta; \alpha_1, \theta) & \text{if } t \in [0, \theta^{\frac{1}{\eta}}) \\ c[F_1(\theta; \alpha_1, \theta) + F_2(t^\eta; \alpha_2, \theta) - F_2(\theta; \alpha_1, \theta)] & \text{if } t \in [\theta^{\frac{1}{\eta}}, \infty). \end{cases} \quad (7)$$

The corresponding survival function of $T$ can then be expressed as follows:

$$S_T(t; \alpha_1, \alpha_2, \theta) = \begin{cases} 1 - cF_1(t^\eta; \alpha_1, \theta) & \text{if } t \in [0, \theta^{\frac{1}{\eta}}) \\ 1 - c[F_1(\theta; \alpha_1, \theta) + F_2(t^\eta; \alpha_2, \theta) - F_2(\theta; \alpha_1, \theta)] & \text{if } t \in [\theta^{\frac{1}{\eta}}, \infty). \end{cases} \quad (8)$$

### 2.2.3. Hazard Function

The hazard function of a random variable $T$ is defined as follows:

$$h_T(t) = \lim_{\Delta t \to \infty} \frac{S_T(t + \Delta t) - S_T(t)}{\Delta t \cdot S_T(t)}.$$

Given the assumption that $T$ is an exponentiated composite random variable induced by the composite random variable $Y$, the hazard function of $T$ can be represented as follows, with the pdf and CDF of its parent composite random variable $Y$:

$$\begin{aligned} h_T(t) &= \frac{f_T(t)}{S_T(t)} = \frac{f_Y(t^\eta)\eta t^{\eta - 1}}{1 - F_Y(t^\eta)}. \\ &= \begin{cases} \frac{cf_1(t^\eta; \alpha_1, \theta)\eta t^{\eta - 1}}{1 - cF_1(t^\eta; \alpha_1, \theta)} & \text{if } t \in [0, \theta^{\frac{1}{\eta}}) \\ \frac{cf_2(t^\eta; \alpha_2, \theta)\eta t^{\eta - 1}}{1 - c[F_1(\theta; \alpha_1, \theta) + F_2(t^\eta; \alpha_2, \theta) - F_2(\theta; \alpha_1, \theta)]} & \text{if } t \in [\theta^{\frac{1}{\eta}}, \infty). \end{cases} \end{aligned} \quad (9)$$

### 2.2.4. Quantile Function

The quantile function of a random variable $Y$ is defined as follows:

$$Q_Y(p) = \inf\{y \in \mathbb{R} : p \leq F_Y(y)\},$$

where $p \in (0, 1)$.

Thus, if $Y$ is a composite random variable and $T$ is the exponentiated composite random variable induced by $Y$, then the quantile function of $T$ can be represented as follows:

$$\begin{aligned} Q_T(p) &= \inf\{t \in \mathbb{R} : p \leq F_T(t)\} \\ &= \inf\{t \in \mathbb{R} : p \leq F_Y(t^\eta)\}. \end{aligned}$$

If $F_Y(y)$ is strictly increasing and continuous, the quantile function $Q_Y(p)$ is then the inverse function of $F_Y(y)$:

$$\begin{aligned} Q_Y(p) &= F_Y^{-1}(p) \\ &= \{y \in \mathbb{R} : F_Y(y) = p\}. \end{aligned}$$

Correspondingly, the quantile function of $T$ can be written as follows in terms of the pdf of its parent composite random variable $Y$:

$$\begin{aligned} Q_T(p) &= F_T^{-1}(p) \\ &= \{t \in \mathbb{R} : F_T(t) = p\} \\ &= \{t \in \mathbb{R} : F_Y(t^\eta) = p\}. \end{aligned} \quad (10)$$

### 3. Special Model: Exponentiated Exponential-Pareto Distribution (EEP)

The exponential-Pareto (EP) composite distribution was first proposed by Teodorescu and Vernic in 2006 [17]. Suppose that $Y$ is a EP random variable. Then, the pdf of $Y$ is first defined as follows:

$$f_Y(y; \lambda, \theta) = \begin{cases} c\lambda e^{-\lambda y} & 0 \le y < \theta \\ c\alpha \frac{\theta^\alpha}{y^{\alpha+1}} & y \ge \theta, \end{cases}$$

Assuming that the continuity and differentiability conditions hold, we have the following:

$$\begin{cases} \lim_{y \to \theta^-} f_Y(y; \lambda, \theta) = \lim_{y \to \theta^+} f_Y(y; \alpha_1, \alpha_2, \theta) \\ \lim_{y \to \theta^-} \frac{df_Y(y; \lambda, \theta)}{dy} = \lim_{y \to \theta^+} \frac{df_Y(y; \lambda, \theta)}{dy}. \end{cases}$$

This generates a system of equations in terms of $\theta, \alpha,$ and $\lambda$:

$$\begin{cases} \lambda e^{-\lambda\theta} = \alpha\theta^{-1} \\ -\lambda^2 e^{-\lambda\theta} = -\alpha(\alpha+1)\theta^{-2}. \end{cases}$$

By simple calculation, $\lambda$ can be expressed in terms of $\alpha$ and $\theta$, as follows:

$$\lambda = \frac{\alpha+1}{\theta}.$$

Additionally, by utilizing the system of equations, the value of $\alpha$ can be determined as the solution of the following equation:

$$(\alpha+1)e^{-(\alpha+1)} = \alpha.$$

Since the shape parameter $\alpha$ of a Pareto distribution is restricted to being positive, the above equation gives a unique solution:

$$\alpha = 0.349976.$$

Since we assume that $f_Y(y; \lambda, \theta)$ is a valid pdf, we can also obtain the unique solution for the normalizing constant $c$ as follows:

$$c = \frac{1}{2 - e^{-(\alpha+1)}} = 0.574464.$$

Thus, the pdf of $Y$ with one parameter $\theta$ can be written as follows:

$$f_Y(y; \theta) = \begin{cases} c(\frac{\alpha+1}{\theta})e^{-\frac{(\alpha+1)y}{\theta}} & y < \theta \\ c\alpha \frac{\theta^\alpha}{y^{\alpha+1}} & y \ge \theta, \end{cases}$$

where $c = 0.574464, \alpha = 0.349976$.

With the power transformation $T = Y^{\frac{1}{\eta}}$, $T$ is obtained as an exponentiated exponential-Pareto (EEP) composite random variable. The pdf of $T$ can be expressed as follows:

$$f_T(t; \theta, \eta) = \begin{cases} c(\frac{\alpha+1}{\theta})e^{-\frac{(\alpha+1)t^\eta}{\theta}}\eta t^{\eta-1} & 0 < t < \theta^{\frac{1}{\eta}} \\ c\alpha \frac{\theta^\alpha}{(t^\eta)^{\alpha+1}}\eta t^{\eta-1} & t \ge \theta^{\frac{1}{\eta}}, \end{cases} \tag{11}$$

Figure 1 shows the pdf of EEP distributions with different parameters. Notice that, when $\eta > 1$, the pdf of EEP distribution is hump-shaped; we discuss this in detail in the next subsection.

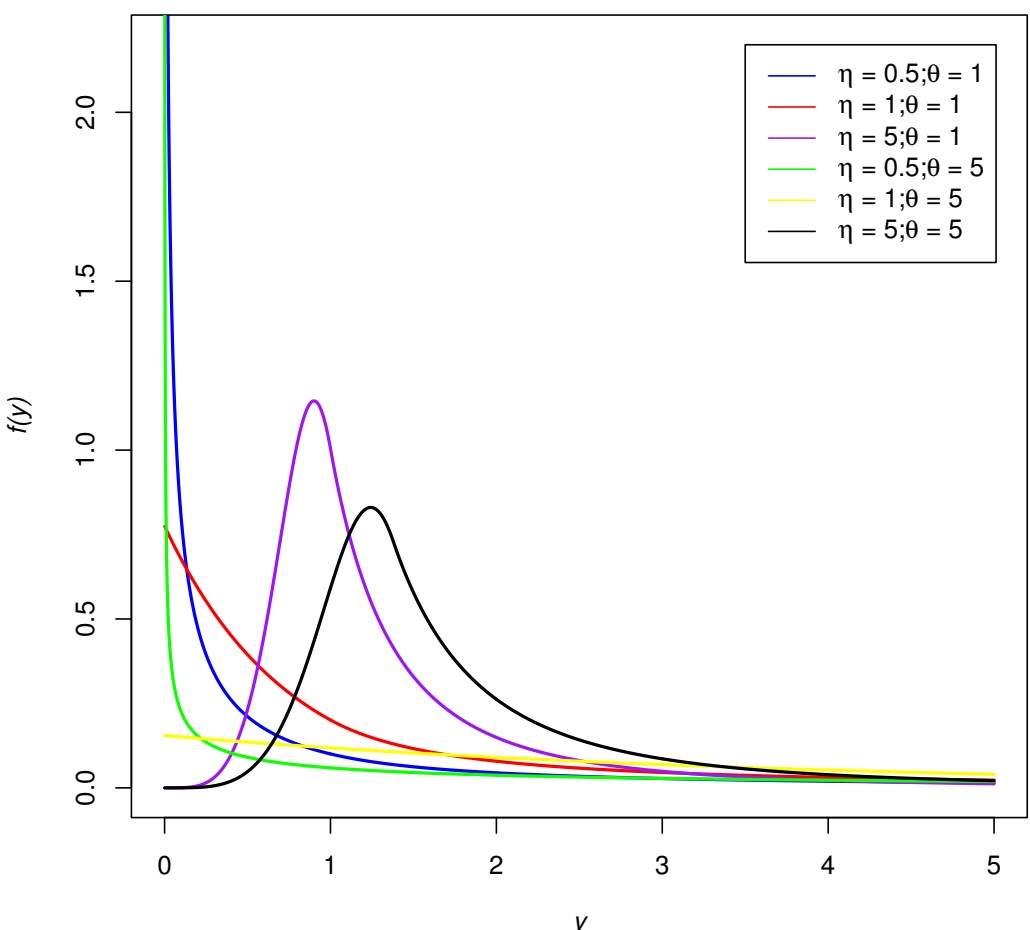

**Figure 1.** The pdf of an EEP distribution with different parameters.

### 3.1. Mode

Since both exponential distribution and Pareto distribution are associated with monotonically decreasing pdf, the pdf of the original EP distribution is also monotonically decreasing. Thus, as an composite distribution, EP still does not have the ability to model the data with hump-shaped frequency distributions. One of the advantages of the EEP distribution against the original EP distribution is that the EEP can have a pdf with a hump shape, or equivalently, the EEP distribution has a mode under certain conditions. We explore the conditions for such hump-shaped distributions in the following part:

**Proposition 1.** *Suppose that $T$ is an EEP composite random variable. Assume that $f_T(t; \theta, \eta)$ is the pdf associated with $T$. Then, the mode of $T$ exists if and only if $\eta > 1$. When $\eta > 1$, the mode of $T$ is $\left[\frac{\theta(\eta-1)}{\eta(\alpha+1)}\right]^{\frac{1}{\eta}}$.*

**Proof of Proposition 1.** First, suppose $t \geq \theta^{\frac{1}{\eta}}$; then, $f_T(t; \theta, \eta) = c\alpha \frac{\theta^\alpha}{(t^\eta)^{\alpha+1}} \eta t^{\eta-1}$.

Correspondingly, $\frac{df_T(t;\theta,\eta)}{dt} = (-\alpha\eta - 1)c\alpha\theta^\alpha \eta t^{-\alpha\eta-2} < 0$ for all $t \geq \theta^{\frac{1}{\eta}}$.

Therefore, when $t \geq \theta^{\frac{1}{\eta}}$, $f_T(t; \theta, \eta)$ is monotonically decreasing.

Since $f_T(t; \theta, \eta)$ is continuous, this implies that the maximum of $f_T(t; \theta, \eta)$ must occur when $t < \theta^{\frac{1}{\eta}}$ if the maximum exists.

Assume $t < \theta^{\frac{1}{\eta}}$, then $f_T(t; \theta, \eta) = c\left(\frac{\alpha+1}{\theta}\right)e^{-\frac{(\alpha+1)t^\eta}{\theta}} \eta t^{\eta-1}$.

When $f_T(t; \theta, \eta)$ is a positive real-valued function, maximizing $f_T(t; \theta, \eta)$ is equivalent to maximizing $\ln f_T(t; \theta, \eta)$. Thus, we use $\ln f_T(t; \theta, \eta)$ for the rest of the derivations.

Then, $\ln f_T(t; \theta, \eta) = \ln(c) + \ln\left(\frac{\alpha+1}{\theta}\right) - \frac{(\alpha+1)t^\eta}{\theta} + \ln(\eta) + (\eta-1)\ln(t)$.

Correspondingly, $\frac{d \ln f_T(t;\theta,\eta)}{dt} = -\frac{(\alpha+1)\eta t^{\eta-1}}{\theta} + \frac{\eta-1}{t}$

Let $\frac{d \ln f_T(t;\theta,\eta)}{dt} = 0$. Then, we have the equation of $t$: $\eta - 1 = \eta(\frac{\alpha+1}{\theta})t^{\eta}$.

Notice that $t > 0$. Hence, the above equation has a solution only when $\eta > 1$. Therefore, the mode of $T$ exists if and only if $\eta > 1$.

When $\eta \geq 1$, the equation has a unique solution: $t = [\frac{\theta(\eta-1)}{\eta(\alpha+1)}]^{\frac{1}{\eta}}$. It can be shown that $\frac{d^2 f_T(t;\theta,\eta)}{dt^2} < 0$ when $t = [\frac{\theta(\eta-1)}{\eta(\alpha+1)}]^{\frac{1}{\eta}}$. Thus, $t = [\frac{\theta(\eta-1)}{\eta(\alpha+1)}]^{\frac{1}{\eta}}$ is the mode of $T$.

□

Therefore, the addition of the exponent parameter $\eta$ in the EEP model definitely improves the flexibility of the original EP model since it can model the data with a hump-shaped distribution.

### 3.2. Moments

The $k$-th moments of the EEP distribution was derived by Liu and Ananda [13] as follows:

$$
\begin{aligned}
E(T^k) = c(\frac{\theta}{\alpha+1})^{\frac{k}{\eta}} & \left[ \Gamma(\frac{k}{\eta}+1) - \Gamma(\frac{k}{\eta}+1, \alpha+1) \right] \\
& + \frac{c\alpha\theta^{\frac{k}{\eta}}}{\alpha - \frac{k}{\eta}},
\end{aligned}
\tag{12}
$$

where $\Gamma(\alpha)$ represents the Gamma function, namely $\Gamma(\alpha) = \int_0^\infty u^{\alpha-1}e^{-u}du$. $\Gamma(.,.)$ When $\frac{k}{\eta} < \alpha$, the $k$-th moment is finite. Correspondingly, the mean of the EEP distribution is as follows:

$$
\begin{aligned}
E(T) = c(\frac{\theta}{\alpha+1})^{\frac{1}{\eta}} & \left[ \Gamma(\frac{1}{\eta}+1) - \Gamma(\frac{1}{\eta}+1, \alpha+1) \right] \\
& + \frac{c\alpha\theta^{\frac{1}{\eta}}}{\alpha - \frac{1}{\eta}}.
\end{aligned}
$$

The mean of the EEP distribution is plotted in Figure 2. The figure shows that the mean is an increasing function of $\theta$ and a decreasing function of $\eta$ for selected $\theta$ and $\eta$ values.

### 3.3. Survival Function and Tail Properties

Since (11) is a continuous differentiable pdf, the cumulative distribution function (CDF) can be derived. By the definition of CDF, we have the following:

$$
F_T(t;\theta,\eta) = \begin{cases} c\left(1 - e^{-\frac{(\alpha+1)t^\eta}{\theta}}\right) & t < \theta^{\frac{1}{\eta}} \\ c\left(2 - e^{-(\alpha+1)} - \frac{\theta^\alpha}{t^{\eta\alpha}}\right) & t \geq \theta^{\frac{1}{\eta}}. \end{cases}
\tag{13}
$$

Notice that the development of EP distribution guarantees that $c(2 - e^{-(\alpha+1)}) = 1$. Therefore,

$$
F_T(t;\theta,\eta) = \begin{cases} c\left(1 - e^{-\frac{(\alpha+1)t^\eta}{\theta}}\right) & t < \theta^{\frac{1}{\eta}} \\ 1 - c\left(\frac{\theta^\alpha}{t^{\eta\alpha}}\right) & t \geq \theta^{\frac{1}{\eta}}. \end{cases}
\tag{14}
$$

Since $T$ is a continuous random variable, the survival function of $T$ is provided correspondingly as follows:

$$
S_T(t;\theta,\eta) = 1 - F_T(t;\theta,\eta),
$$

that is,

$$S_T(t;\theta,\eta) = \begin{cases} 1 - c\left(1 - e^{-\frac{(\alpha+1)t^\eta}{\theta}}\right) & t < \theta^{\frac{1}{\eta}} \\ c\left(\frac{\theta^\alpha}{t^{\eta\alpha}}\right) & t \geq \theta^{\frac{1}{\eta}}. \end{cases} \tag{15}$$

Now, given that we derived the formula of CDF and the survival function of $T$, we can further discuss the tail properties of the EEP distribution.

**Proposition 2.** *If $T$ follows an EEP distribution with a CDF F, then the distribution of T has a heavy tail.*

**Proof of Proposition 2.** We only prove that $\lim_{t\to\infty} e^{\lambda t} S_T(t) = \lim_{t\to\infty}\left(\frac{c\theta^\alpha e^{\lambda t}}{t^{\eta\alpha}}\right) = \infty$ for all $\lambda > 0$.

Notice $\eta\alpha > 0$ based on our assumption. Denote $\lceil\eta\alpha\rceil$ as the smallest integer greater than $\eta\alpha$.

Then, since $\lim_{t\to\infty} c\theta^\alpha e^{\lambda t} = \infty$ and $\lim_{t\to\infty} t^{\eta\alpha} = \infty$, apply L'Hospital rule $\lceil\eta\alpha\rceil$ times. We have $\lim_{t\to\infty} e^{\lambda t} S_T(t) = \lim_{t\to\infty}\left(\frac{c\lambda^{\lceil\eta\alpha\rceil}\theta^\alpha e^{\lambda t}}{\prod_{i=1}^{\lceil\eta\alpha\rceil}(\eta\alpha-i)t^{\eta\alpha-\lceil\eta\alpha\rceil}}\right) = \infty$.

Therefore, the distribution of $T$ has a heavy tail.

□

Furthermore, we could also specify the conditions that an EEP distribution has a fat tail. Suppose that the survival function of a random variable $X$ satisfies the following:

$$S(x) \sim x^{-\beta} \text{ as } x \to \infty, \alpha > 0$$

Notice that a distribution is said to have a fat tail when the tail index $\beta < 2$. Notice that, for an EEP random variable $T$, $S_T(t) \sim t^{-\eta\alpha}$. Therefore, only when $\eta\alpha < 2$, the EEP distribution is considered to have a fat tail.

*3.4. Hazard Function*

For a continuous random variable $T$, the hazard function is defined as follows:

$$h_T(t) = \frac{f_T(t)}{S_T(t)}.$$

Therefore, by utilizing the pdf and survival function of $Y$, the hazard function of a EEP random variable $T$ can be derived as follows:

$$h_T(t;\theta,\eta) = \begin{cases} \dfrac{c\left(\frac{\alpha+1}{\theta}\right)e^{-\frac{(\alpha+1)t^\eta}{\theta}}\eta t^{\eta-1}}{1-c+ce^{-\frac{(\alpha+1)t^\eta}{\theta}}} & t < \theta^{\frac{1}{\eta}} \\ \\ \alpha\eta t^{-1} & t \geq \theta^{\frac{1}{\eta}} \end{cases} \tag{16}$$

Notice that $h_T(t;\theta,\eta)$ is monotonically decreasing when $t \geq \theta^{\frac{1}{\eta}}$. For some selections of $\theta$ and $\eta$ values, the hazard function $h_T(t : \theta,\eta)$ could be UBT. Figure 3 shows the hazard functions of EEP distributions with different parameters.

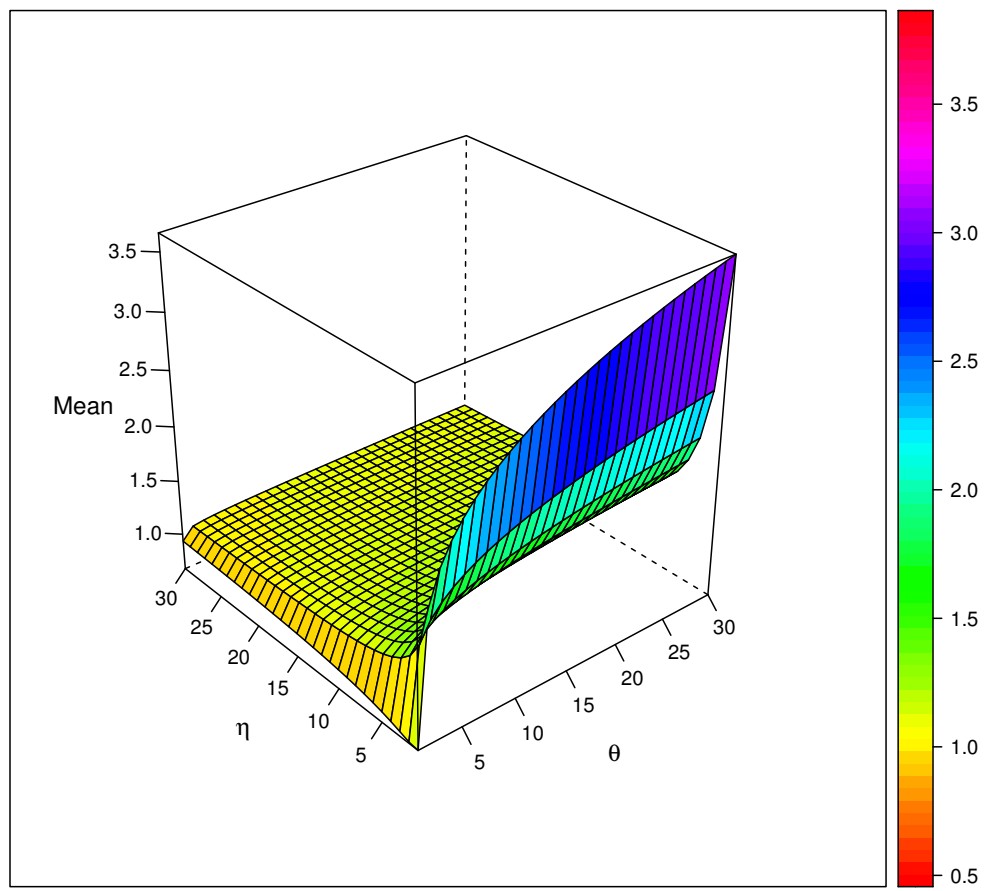

**Figure 2.** Mean plot of an EEP distribution with different parameters.

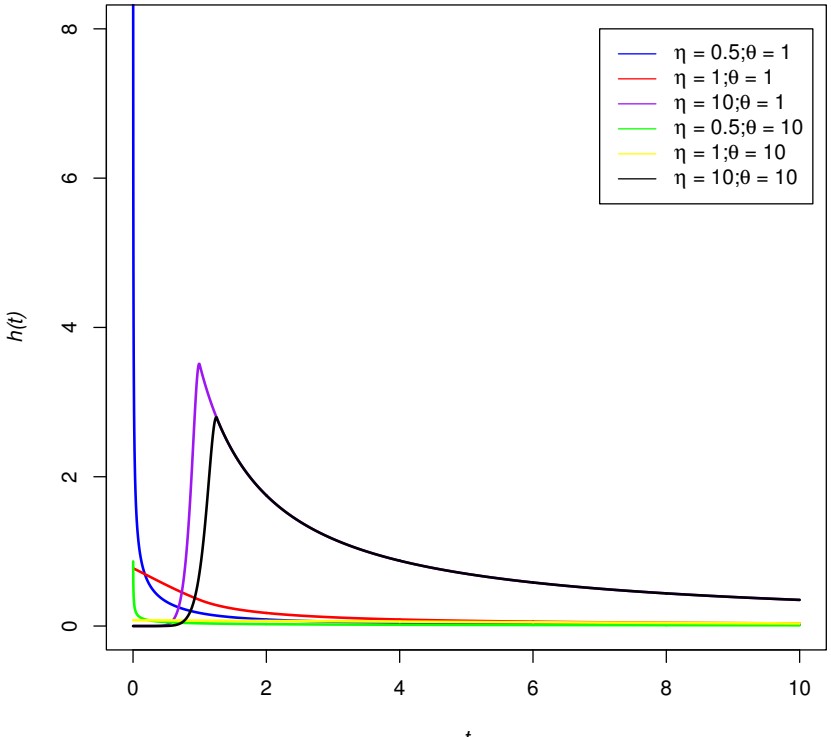

**Figure 3.** Hazard function of an EEP distribution with different parameters.

### 3.5. Quantile Function

The quantile function of a continuous random variable $T$ is defined as the inverse function of its CDF $F_T$. Since the explicit form of $F_T$ is derived in (10), we have the quantile function an EEP random variable $T$ as follows:

$$F_T^{-1}(p;\theta,\eta) = \begin{cases} [\ln(\frac{c}{c-p})\frac{\theta}{\alpha+1}]^{\frac{1}{\eta}} & p < c[1 - e^{-(\alpha+1)}] \\[3mm] \left(\frac{c}{1-p}\right)^{\frac{1}{\alpha\eta}}\theta^{\frac{1}{\eta}} & p \geq c[1 - e^{-(\alpha+1)}]. \end{cases} \tag{17}$$

In fact, given that the values of $c$ and $\alpha$ are fixed, the value of $c[1 - e^{-(\alpha+1)}]$ could be obtained. Notice that $c[1 - e^{-(\alpha+1)}] < 0.5$. Hence, the median (0.5 quantile) of an EEP random variable could be expressed as follows, in terms of $\theta$ and $\eta$:

$$M(\theta,\eta) = F_T^{-1}(p = 0.5;\theta,\eta)$$
$$= (2c)^{\frac{1}{\alpha\eta}}\theta^{\frac{1}{\eta}}$$

The median of EEP distribution is plotted in Figure 4. Similar to what we observed in Figure 2, the median of an EEP distribution is an increasing function in $\theta$ and a decreasing function in $\eta$.

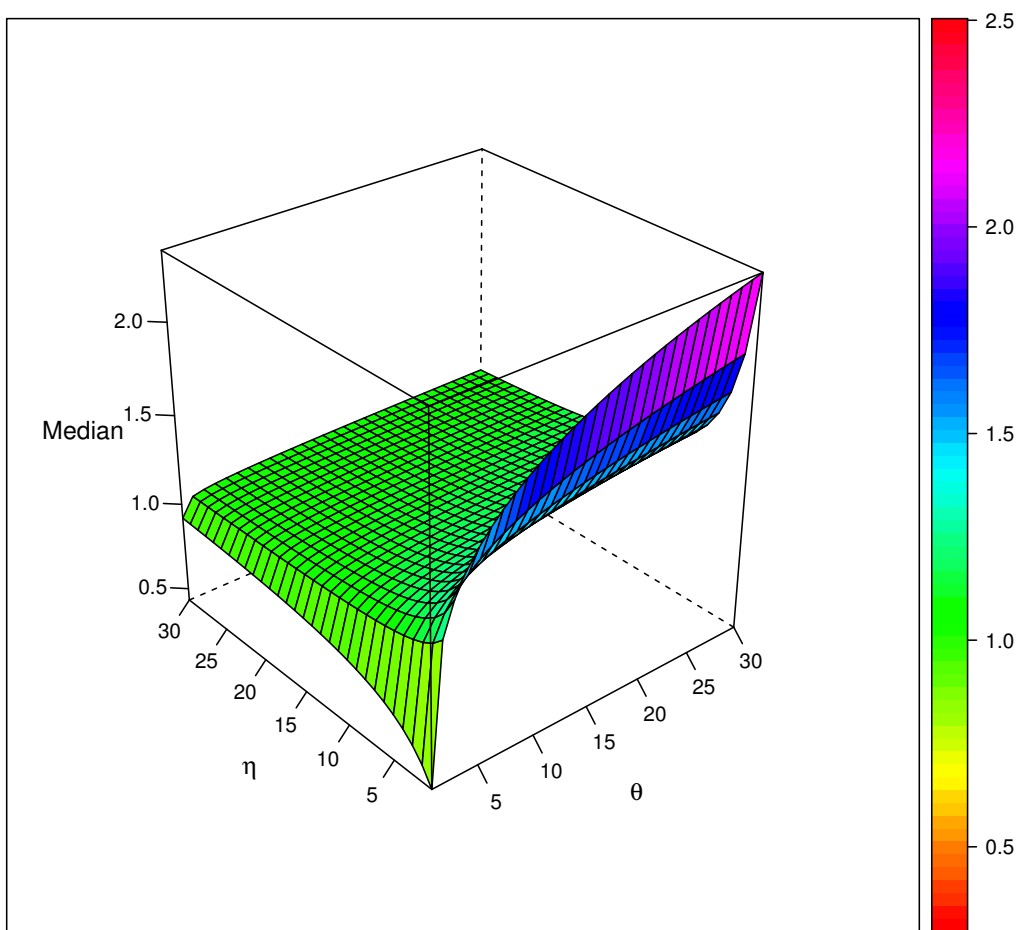

**Figure 4.** Median plot of EEP distribution with different parameters.

*3.6. Median Residual Lifetime*

For a random variable $T$, the median residual life time is denoted as $\mathrm{MERL}(t)$:

$$\frac{S_T(t + \mathrm{MERL}(t))}{S_T(t)} = \frac{1}{2}, \tag{18}$$

where $S_T(t)$ is the survival function of $T$. Therefore, by utilizing Equation (15), we have the following:

$$\mathrm{MERL}_T(t|\theta, \eta) = \begin{cases} \left( \frac{2c\theta^\alpha}{1 - c - ce^{-\frac{\alpha+1}{\theta} t^\eta}} \right)^{\frac{1}{\alpha\eta}} - t & t < \theta^{\frac{1}{\eta}} \\ 2^{\frac{1}{\eta\alpha}} t - t & t \geq \theta^{\frac{1}{\eta}}. \end{cases} \tag{19}$$

MERL is a very important dynamic measure for distributions with heavy tails. The median residual lifetime functions for selected values of parameters are illustrated in Figure 5. Essentially, if two EEP models have the same value for the parameter $\eta$, they will have the same median residual lifetime if $t$ is large.

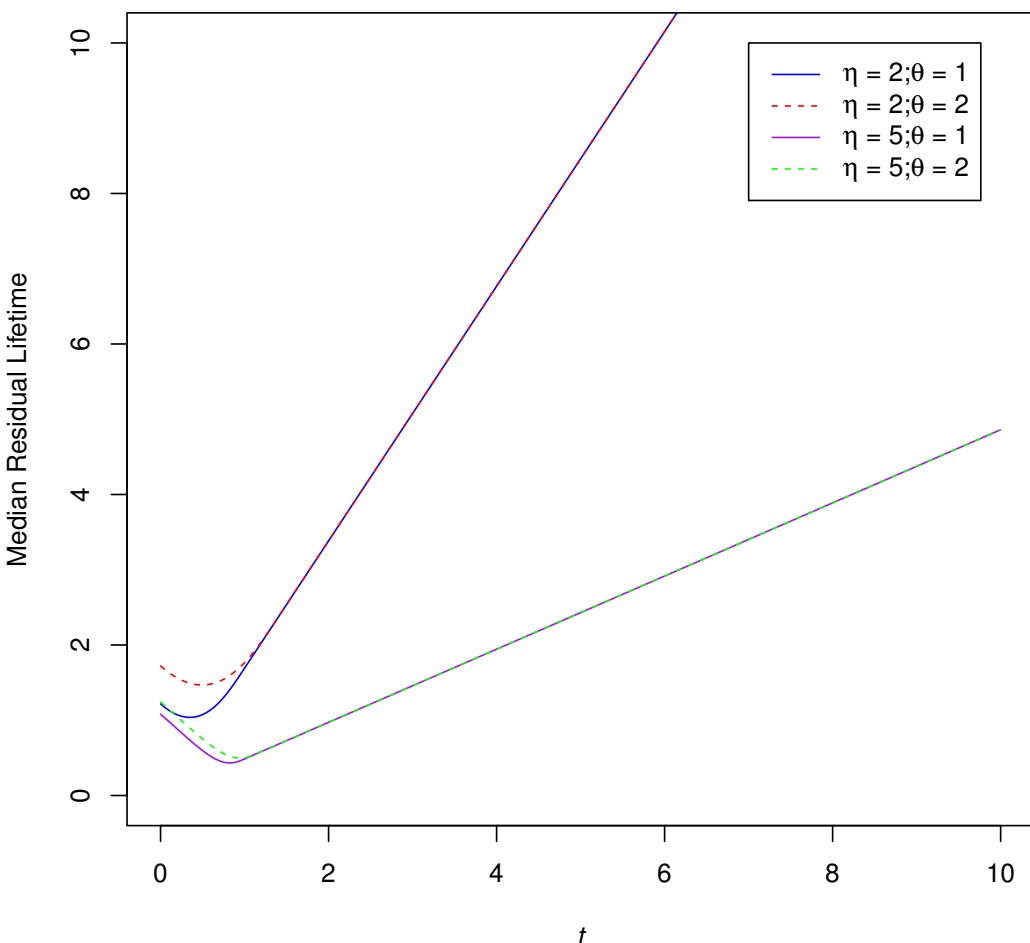

**Figure 5.** The median residual lifetime function plot of EEP distribution with different parameters.

## 4. Parameter Estimation

*Parameter Search for Two-Parameter EEP Distributions*

Suppose that $t = (t_1, t_2, \ldots, t_n)$ is an independent identically distributed (i.i.d.) EEP sample with parameters $\theta$ and $\eta$. Without loss of generality, assume $t_1 < t_2 < \ldots < t_n$.

Assume that there exists a positive integer $m$ such that $t_m < \theta^{\frac{1}{\eta}} < t_{m+1}$. The corresponding log-likelihood of $t$ can be written as follows:

$$
\begin{aligned}
l(\theta, \eta; t) = &\; n\ln(c) + n\ln(\eta) + m\ln(\alpha + 1) \\
&+ (n - m)\ln(\alpha) + (\eta - 1)\sum_{i=1}^{m}\ln(t_i) \\
&- (\alpha\eta + 1)\sum_{j=m+1}^{n}\ln(t_j) + \alpha n\ln(\theta) \\
&- [(\alpha + 1)m]\ln(\theta) - \frac{\alpha + 1}{\theta}\sum_{i=1}^{m}t_i^{\eta}
\end{aligned}
\tag{20}
$$

Due to the existence of the location parameters ($\theta$ and $\eta$) in the EEP model, the maximum likelihood estimates (MLE) of the model parameters cannot be obtained directly by maximizing the log-likelihood function $l(\theta, \eta; t)$. Liu and Ananda [13,14] introduced a grid-search algorithm to find the estimates of the model parameters. In fact, the estimates can also be obtained by utilizing the 'optim' function in R. The steps for finding MLE of the model parameters are as follows:

1. Arrange the observations in the sample in increasing order such that $t_{(1)} \leq t_{(2)} \leq \ldots \leq t_{(n)}$.
2. For $m = 1, 2, \ldots, n - 1$, maximize the objective function $l(\theta, \eta; t)$ and obtain $(\hat{\theta}_1, \hat{\eta}_1), (\hat{\theta}_2, \hat{\eta}_2)$, ..., $(\hat{\theta}_{n-1}, \hat{\eta}_{n-1})$ correspondingly.
3. Start from $m = 1$, check the condition $t_{(1)} < (\hat{\theta}_1)^{\frac{1}{\hat{\eta}_1}} < t_{(2)}$. If this is true, then $\hat{\theta}_1$ and $\hat{\eta}_1$ are the estimates for $\theta$ and $\eta$. If not, go to the next step.
4. For $m = 2$, check the condition $t_{(2)} < (\hat{\theta}_2)^{\frac{1}{\hat{\eta}_2}} < t_{(3)}$. If this is true, then $\hat{\theta}_2$ and $\hat{\eta}_2$ are the estimates for $\theta$ and $\eta$. Repeat this procedure until the correct $m$ is detected. With the correct $m$, the corresponding $\hat{\theta}_m$ and $\hat{\eta}_m$ are the estimates for $\theta$ and $\eta$.

The above algorithm is used to compute the parameter estimates of EEP in the next section.

## 5. Goodness-of-Fit Tests with Simulation Studies

In this section, we introduce two goodness-of-fit (GoF) tests to assess the GoF of the EEP distribution against its parent distribution, EP distribution. The corresponding hypothesis to be tested is as follows:

$$
\begin{cases}
H_0 : \eta = 1 & \text{i.e., data follow an EP}(\theta) \\
H_1 : \eta \neq 1 & \text{i.e., data follow an EEP}(\theta, \eta).
\end{cases}
\tag{21}
$$

The MLE of $\theta$ and $\eta$ of an EEP distribution can be obtained numerically using the proposed algorithm. Thus, the likelihood ratio test (LRT) could be utilized to test the above hypothesis. Assume that the likelihood function $l(\theta, \eta | t)$ takes the form in (17). The LRT statistic $\lambda$ for the given hypothesis is as follows:

$$
\lambda = 2\sup_{\theta \in \mathbb{R}^+} l(\theta, \eta = 1; t) - 2\sup_{\theta \in \mathbb{R}^+, \eta \in \mathbb{R}^+} l(\theta, \eta | t),
\tag{22}
$$

where $\lambda$ follows a $\chi^2$ distribution with a degree of freedom 1.

As an alternative procedure, the asymptotic Wald's test could also be used to test the hypothesis, given the Fisher information matrix and MLE of $\theta$ and $\eta$. Consider $\Theta = (\theta, \eta)$. The Fisher information matrix was derived by Liu and Ananda [13,14] as follows:

$$I(\Theta) = \begin{pmatrix} -\frac{\partial^2 l(\theta,\eta;t)}{\partial \theta^2} & -\frac{\partial^2 l(\theta,\eta);t}{\partial \theta \partial \eta} \\ -\frac{\partial^2 l(\theta,\eta;t)}{\partial \theta \partial \eta} & -\frac{\partial^2 l(\theta,\eta;t)}{\partial \eta^2} \end{pmatrix}$$

$$= \begin{pmatrix} I_{11} & I_{12} \\ I_{21} & I_{22} \end{pmatrix},$$

where

$$\begin{cases} I_{11} = \frac{\alpha n - (\alpha+1)m}{\theta^2} + \frac{2(\alpha+1)}{\theta^3} \sum_{i=1}^{m} t_i^{\eta} \\ I_{12} = -\frac{\alpha+1}{\theta^2} \sum_{i=1}^{m} t_i^{\eta} \ln(t_i) \\ I_{21} = -\frac{\alpha+1}{\theta^2} \sum_{i=1}^{m} t_i^{\eta} \ln(t_i) \\ I_{22} = \frac{n}{\eta^2} + \frac{\alpha+1}{\theta} \sum_{i=1}^{m} t_i^{\eta} [\ln(t_i)]^2. \end{cases}$$

Essentially, the expected value $E(I(\theta))$ is the Fisher information matrix $\mathcal{I}(\Theta)$.

Consider $\hat{\Theta} = (\hat{\theta}, \hat{\eta})$ as the MLE for $\Theta = (\theta, \eta)$. By the asymptotic property of MLE, we have the following under appropriate regularity conditions:

$$(\hat{\Theta} - \Theta) \xrightarrow{\mathcal{D}} \mathcal{N}(\mathbf{0}_2, \mathcal{I}^{-1}(\Theta)), \text{ as } n \to \infty,$$

where $\mathcal{I}(\Theta)$ stands for the Fisher information matrix of $\Theta$, $\mathcal{D}$ represents convergence in distribution, and $\mathcal{N}(\mathbf{0}_2, \mathcal{I}^{-1}(\Theta))$ is the bivariate normal distribution with covariance matrix $\mathcal{I}^{-1}(\Theta)$.

However, in reality, $\mathcal{I}(\Theta) = E(I(\theta))$ cannot be easily evaluated. Therefore, the observed Fisher information matrix $\mathcal{I}(\hat{\Theta})$ is used as a substitute.

$$\mathcal{I}(\hat{\Theta}) = \begin{pmatrix} \hat{\mathcal{I}}_{11} & \hat{\mathcal{I}}_{12} \\ \hat{\mathcal{I}}_{21} & \hat{\mathcal{I}}_{22} \end{pmatrix},$$

where,

$$\begin{cases} \hat{\mathcal{I}}_{11} = \frac{\alpha n - (\alpha+1)m}{\hat{\theta}^2} + \frac{2(\alpha+1)}{\hat{\theta}^3} \sum_{i=1}^{m} t_i^{\hat{\eta}} \\ \hat{\mathcal{I}}_{12} = -\frac{\alpha+1}{\hat{\theta}^2} \sum_{i=1}^{m} t_i^{\hat{\eta}} \ln(t_i) \\ \hat{\mathcal{I}}_{21} = -\frac{\alpha+1}{\hat{\theta}^2} \sum_{i=1}^{m} t_i^{\hat{\eta}} \ln(t_i) \\ \hat{\mathcal{I}}_{22} = \frac{n}{\hat{\eta}^2} + \frac{\alpha+1}{\hat{\theta}} \sum_{i=1}^{m} t_i^{\hat{\eta}} [\ln(t_i)]^2. \end{cases} \tag{23}$$

Correspondingly, $\mathcal{I}^{-1}(\hat{\Theta})$ can be obtained as follows:

$$\mathcal{I}^{-1}(\hat{\Theta}) = \frac{1}{\hat{\mathcal{I}}_{11}\hat{\mathcal{I}}_{22} - \hat{\mathcal{I}}_{12}\hat{\mathcal{I}}_{21}} \begin{pmatrix} \hat{\mathcal{I}}_{22} & -\hat{\mathcal{I}}_{12} \\ -\hat{\mathcal{I}}_{21} & \hat{\mathcal{I}}_{11} \end{pmatrix} \tag{24}$$

Since the hypothesis of interest is only involved with the parameter $\eta$, we could carry out the Wald's test procedure easily with the following test statistic:

$$\sqrt{W} = \frac{\hat{\eta} - 1}{\sqrt{\text{Var}(\hat{\eta})}}, \tag{25}$$

where, from (16) and (17), $\text{Var}(\hat{\eta}) = \frac{\hat{\mathcal{I}}_{11}}{\hat{\mathcal{I}}_{11}\hat{\mathcal{I}}_{22} - \hat{\mathcal{I}}_{12}\hat{\mathcal{I}}_{21}}$.

We conducted the simulations to assess the type 1 error performances and the power performances of both tests under different scenarios. For all the simulation scenarios, r = 10,000 samples were generated from the EEP density provided in (11). The R package 'mistr' [25] was utilized to generate the random samples.

To evaluate the Type 1 error rates of the two GoF tests, we designed twelve different simulation scenarios with different $\theta$ values and sample sizes. The details of the simulation

scenarios are listed in Table 1. Table 1 shows the type 1 error performance of both tests under twelve different scenarios. We noticed that both tests can control the type 1 error rate well for all the simulation scenarios.

To assess the power performances of the two GoF tests, we designed various simulation scenarios. The simulation scenarios are described as follows:

- Three different sample sizes: $n = 50, 100, 200$.
- Four different true $\theta$ values: $\theta = 0.01, 0.1, 1, 10$.
- Eleven true $\eta$ values: $\eta = 0.5, 0.6, 0.7, 0.8, 0.9, 1, 1.1, 1.2, 1.3, 1.4, 1.5$.

In total, we generated 132 simulation scenarios.

The power comparison of two tests are provided in Figure 6. It can be observed that the power of the tests drops when $\eta$ increases from 0.5 to 1 and rises when $\eta$ increases from 1 to 1.5. When the sample size becomes larger, the power of the tests increases. It is also noteworthy that, when the sample size is 50 or 100, the power of the LRT was slightly better than Wald's test. When the sample size increases to 200, both tests demonstrated similar performance. However, the value of $\theta$ did not affect the power of both tests significantly, under all the simulation scenarios.

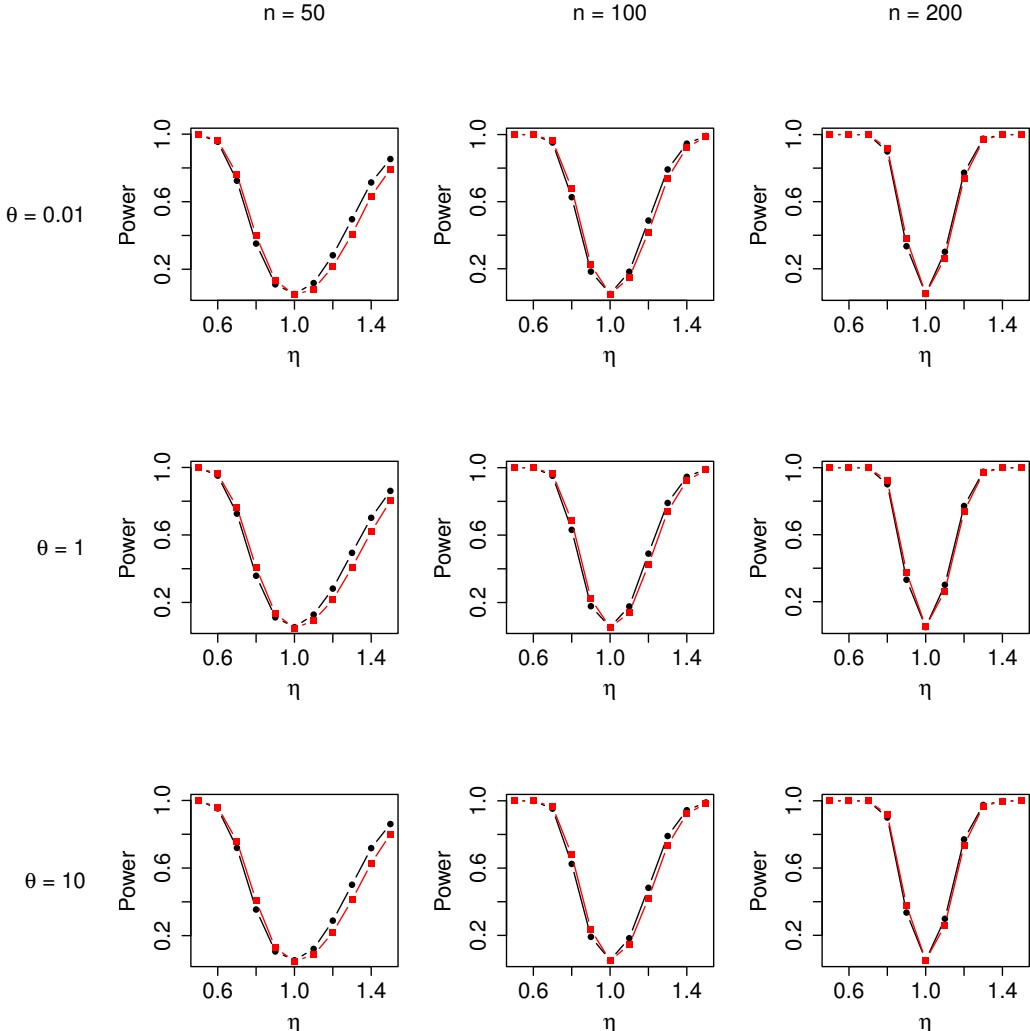

**Figure 6.** Power comparison of the Likelihood ratio test and asymptotic Wald's test (the red line represents the asymptotic Wald's test, and the black line represents the LRT).

**Table 1.** Type 1 error rates for likelihood ratio test and asymptotic Wald's test.

| $\theta$ | Sample Size ($n$) | LRT | | Wald's Test | |
|---|---|---|---|---|---|
| | | $\alpha = 0.01$ | $\alpha = 0.05$ | $\alpha = 0.01$ | $\alpha = 0.05$ |
| 0.01 | $n = 50$ | 0.0110 | 0.0536 | 0.0107 | 0.0484 |
| | $n = 100$ | 0.0125 | 0.0496 | 0.0105 | 0.0477 |
| | $n = 200$ | 0.0095 | 0.0499 | 0.0101 | 0.0503 |
| 0.1 | $n = 50$ | 0.0094 | 0.0514 | 0.0087 | 0.0443 |
| | $n = 100$ | 0.0108 | 0.0529 | 0.0104 | 0.0519 |
| | $n = 200$ | 0.0112 | 0.0495 | 0.0113 | 0.0517 |
| 1 | $n = 50$ | 0.0116 | 0.0537 | 0.0111 | 0.0469 |
| | $n = 100$ | 0.0120 | 0.0542 | 0.0121 | 0.0523 |
| | $n = 200$ | 0.0100 | 0.0484 | 0.0113 | 0.0508 |
| 10 | $n = 50$ | 0.0120 | 0.0544 | 0.0095 | 0.0490 |
| | $n = 100$ | 0.0104 | 0.0523 | 0.0103 | 0.0495 |
| | $n = 200$ | 0.0121 | 0.0518 | 0.0127 | 0.0523 |

## 6. Application to Real Data

In this section, we applied the EEP model to three real data sets to assess its ability in fitting reliability data. For comparison, we choose two models with Pareto tails. The first one is the modified heavy-tailed Pareto model [26]. This is a model that was recently developed to fit the data with UBT-shaped hazard rates. The other one is the Weibull–Pareto composite model [20,23]. This model was previously used to model unimodal survival and reliability data.

We compared the performance of the proposed EEP model against the original EP model, generalized in terms of Akaike information criterion (AIC) [27], AIC3 [28], corrected Akaike information criterion (AICc) [29], and consistent Akaike information criterion (cAIC) [30]. In addition, to justify that the EEP model is proper compared to its parent EP model, LRT and Wald's test were conducted with the hypothesis in (18). In addition, the Kolmogorov–Smirnoff (KS) test was used to assess the goodness-of-fit of the EEP and the EP model.

### 6.1. Second Reactor Pump Data

We utilized a data set that describes the time between failure of secondary reactor pumps [31]. The data set is illustrated in Table 2. It was used by previous researchers [6], and it is considered to be a data set that has a upside-down bathtub-shaped hazard rate. The TTT plot and the box plot are presented in Figure 7. The box plot suggests that the data are heavily right-skewed. The TTT plot suggests that the data are associated with a UBT hazard function.

**Table 2.** Second reactor pump failure time data.

| |
|---|
| 2.160, 0.150, 4.082, 0.746, 0.358, 0.199, 0.402, 0.101, 0.605, 0.954, |
| 1.359, 0.273, 0.491, 3.465, 0.070, 6.560, 1.060, 0.062, 4.992, 0.614, |
| 5.320, 0.347, 1.921 |

Table 3 presented the estimates with corresponding standard errors, LRT test statistic, Wald test statistic, and corresponding *p*-values from both tests. We were able to reject the null hypothesis for both of the tests at the significance level $\alpha = 0.05$. Thus, we concluded that EEP is more proper compared to EP when fitting to the data set. The KS test also suggested that we could not reject the null hypothesis of the data being from an EEP distribution.

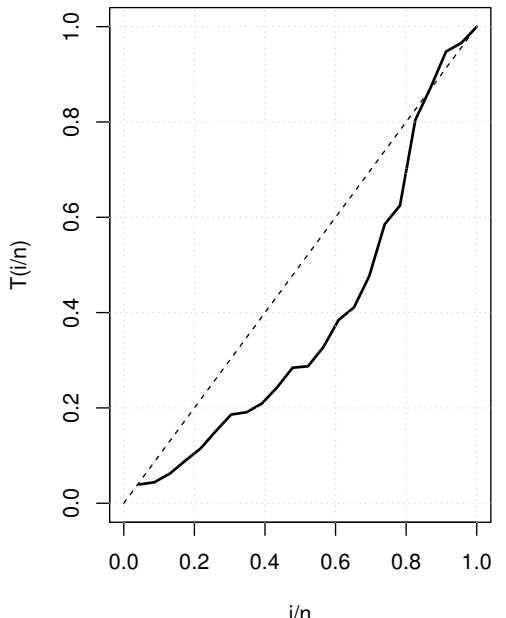
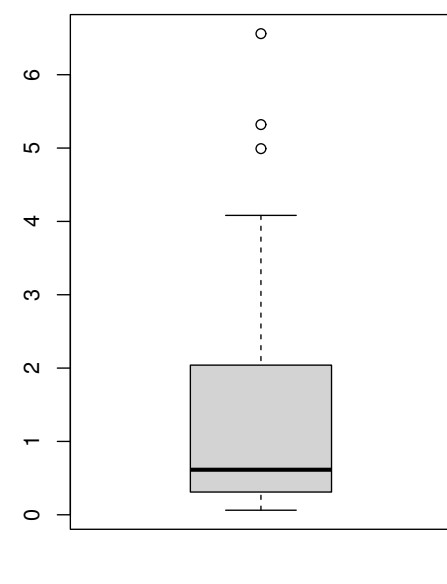

**Figure 7.** TTT plot (**left**) and box plot (**right**) of the second reactor pump data set.

**Table 3.** Summary of the EEP and EP fit to the second reactor pump data set.

| Model | Estimates | LRT Statistic ($p$-Value) | Wald Statistic ($p$-Value) | KS Statistic ($p$-Value) |
|---|---|---|---|---|
| EEP | $\theta = 0.25516$ $\eta = 1.73300$ | 6.32602 (0.01190) | 4.45271 (0.03485) | 0.13043 (0.9924) |
| EP | $\theta = 0.60707$ | - | - | 0.26087 (0.4218) |

A comparisons of the four different models is presented in Table 4. In terms of all GoF measures, the EEP model demonstrated better performances compared to the original EP model. Moreover, the EEP model presented comparable performances against the heavy-tailed Pareto model and the Weibull–Pareto model, with less parameters. This essentially suggests that the EEP model is capable of explaining the data features with only two parameters. Figure 8 contains the plot-fitted EEP hazard function and the histogram with the fitted pdf. The fitted hazard function possesses a UBT shape. Figure 9 demonstrates a comparison of the empirical survival function and the fitted EEP survival function. It can be clearly seen that EEP provided great performance when fitting to the second reactor pump data set.

**Table 4.** Comparison of the four models in terms of the five GoF measures.

| Model | $p$ | AIC | AIC3 | AICc | CAIC |
|---|---|---|---|---|---|
| EEP | 2 | 72.29528 | 74.29528 | 72.89528 | 76.56627 |
| EP | 1 | 76.62130 | 77.62130 | 76.81178 | 78.75679 |
| Generalized Heavy-Tailed Pareto | 3 | 69.86288 | 72.86288 | 71.12604 | 76.26936 |
| Weighted Weibull–Pareto Composite | 4 | 69.71532 | 73.73368 | 71.95590 | 78.27566 |

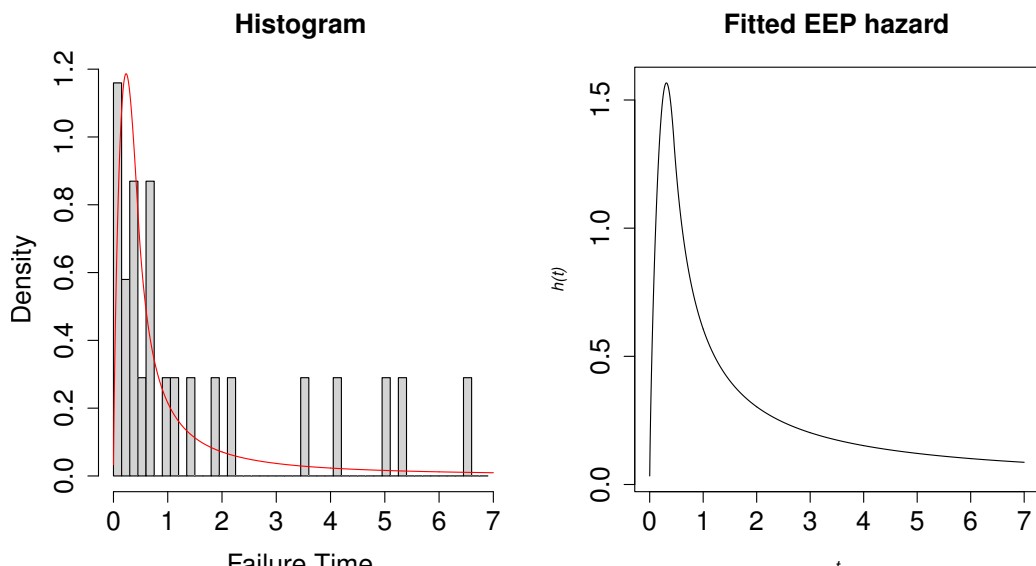

**Figure 8.** Fitted hazard function of EEP distribution for secondary reactor pumps data set.

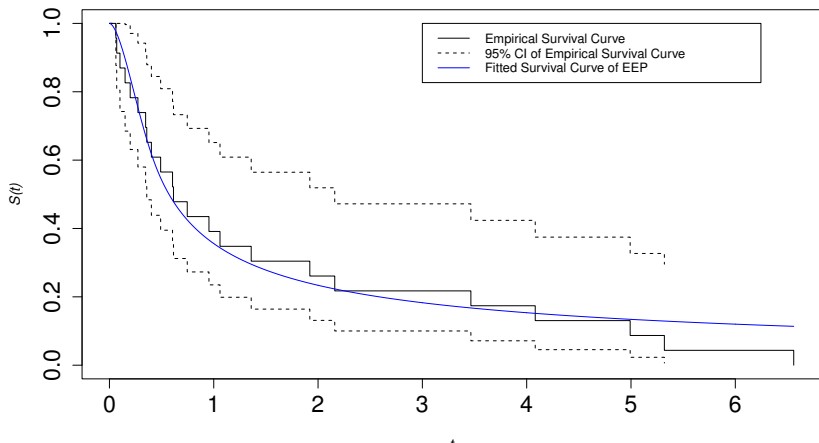

**Figure 9.** Fitted EEP survival curve for secondary reactor pumps data set.

*6.2. Electrical Breakdown of An Insulating Fluid*

We also utilized the time to electrical breakdown of insulating fluid data [32] to assess the performance of the EEP model. In our analysis, we chose the data set with a test voltage 30 kV. Table 5 illustrates the time to electrical breakdown in terms of hours. Figure 10 consists of the TTT plot and the box plot of the data set. The box plot suggests that the data are right-skewed. The TTT plot confirms that the data are associated with a UBT hazard function.

**Table 5.** Time to electric breakdown data.

| |
|---|
| 7.74, 17.05, 20.46, 21.02 22.66, 43.40, |
| 47.30, 139.07 144.12, 175.88, 194.90 |

The estimates, LRT test statistic, Wald test statistic, and corresponding *p*-values from both tests are presented in Table 6. We were able to reject the null hypothesis for both of the tests at the significance level $\alpha = 0.05$. Thus, we concluded that EEP is better compared to EP when fitting to the data set. In addition, the KS test also suggested that we could not reject the null hypothesis of the data being from an EEP distribution at the level of $\alpha = 0.05$, while we rejected the null hypothesis of the data being from an EP distribution.

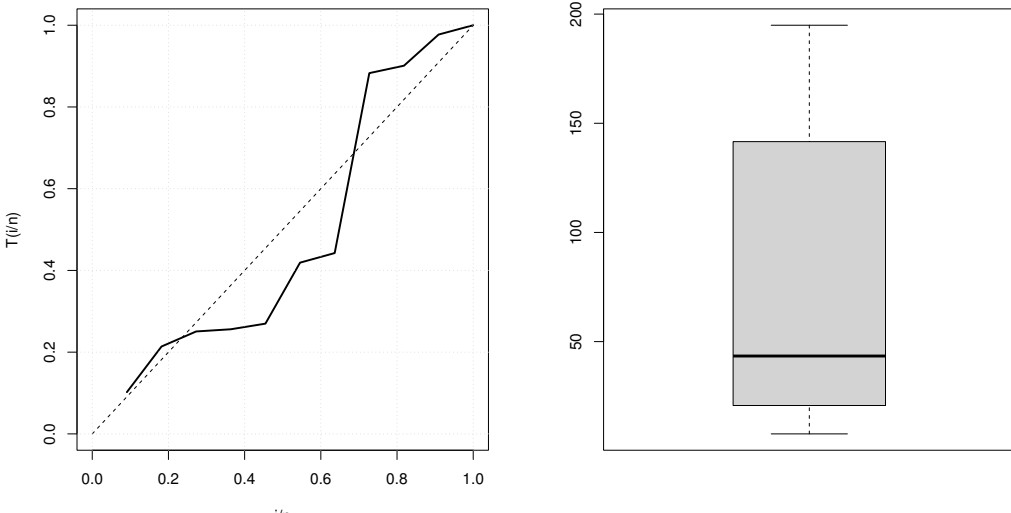

**Figure 10.** TTT plot (**left**) and box plot (**right**) of the time to electric breakdown data set.

**Table 6.** Summary of the EEP and EP fit to the time to electric breakdown set.

| Model | Estimates | LRT Statistic (*p*-Value) | Wald Statistic (*p*-Value) | KS Statistic (*p*-Value) |
|---|---|---|---|---|
| EEP | $\theta = 3326.133$ $\eta = 2.473841$ | 4.08993 (0.04314) | 7.090596 (0.008) | 0.54545 (0.0758) |
| EP | $\theta = 41.39566$ | - | - | 0.90909 (0.0002254) |

Table 7 summarizes the comparison of the four models in terms of all GoF measures. Compared to the original EP model, the EEP model performed better in terms of all the measures. Moreover, the EEP model demonstrated better performance compared to the three-parameter generalized heavy-tailed Pareto model and the four-parameter Weibull–Pareto model. This implies that the EEP model has better abilities to explain the data with only two parameters. Figure 11 consists of the plot-fitted EEP hazard function and the histogram with the fitted pdf. The fitted hazard function has a UBT shape. Figure 12 presents the comparison of the empirical survival function and the fitted EEP survival function.

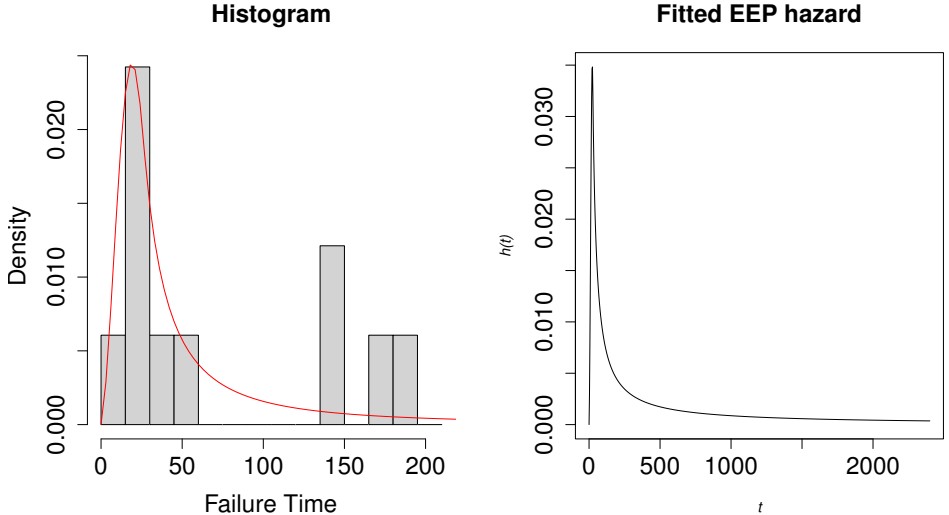

**Figure 11.** The fitted hazard function of EEP distribution for the time to electric breakdown data set.

**Table 7.** Comparison of the four models in terms of the five GoF measures.

| Model | $p$ | AIC | AIC3 | AICc | CAIC |
|---|---|---|---|---|---|
| EEP | 2 | 121.6466 | 123.6466 | 123.1466 | 124.4424 |
| EP | 1 | 126.7372 | 127.7372 | 127.1816 | 128.1351 |
| Generalized Heavy-Tailed Pareto | 3 | 123.4848 | 125.2912 | 125.7197 | 126.4848 |
| Weighted Weibull–Pareto Composite | 4 | 125.2129 | 129.2129 | 131.8795 | 130.8045 |

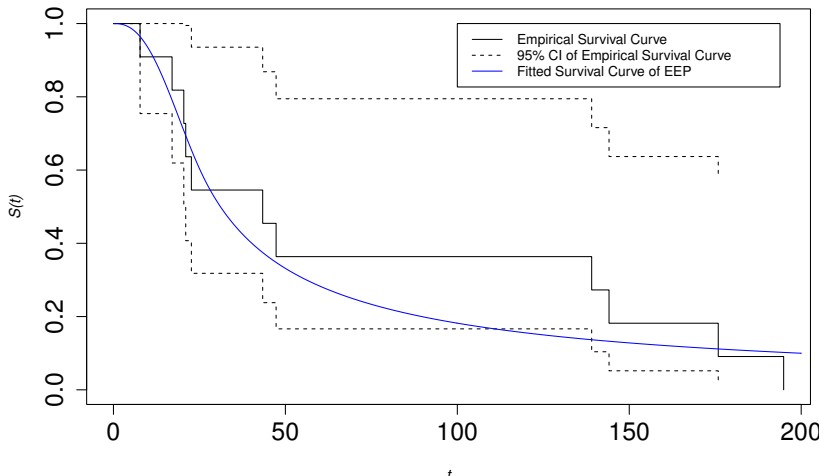

**Figure 12.** Fitted EEP survival curve for time to electric breakdown data set.

## 7. Concluding Remark and Future Work

In this paper, we introduced a special member from the generalized family of exponentiated composite distributions, named exponentiated exponential-Pareto distribution. We derived the properties of this distribution including the moments, the survival function, the hazard function, and the quantile function. We also discussed the parameter estimation and the goodness-of-fit tests for this distribution against its parent distribution. Limited simulations were conducted to compare the performances of the goodness-of-fit tests. In the section with a real data analysis, the proposed distribution demonstrated great performance when fitted to specific reliability data sets with hazard rates being upside-down bathtub-shaped. We hope that this new model can provide different insights into reliability data modeling. Since we only studied one special distribution from the family of generalized exponentiated composite distributions in this paper, the exploration of more special distributions from this family is warranted.

**Author Contributions:** Conceptualization, B.L. and M.M.A.A.; methodology, B.L. and M.M.A.A.; software, B.L.; validation, B.L. and M.M.A.A.; formal analysis, B.L.; investigation, B.L. and M.M.A.A.; resources, B.L.; writing—original draft preparation, B.L.; writing—review and editing, B.L. and M.M.A.A.; supervision, M.M.A.A.; project administration, B.L. and M.M.A.A.; All authors have read and agreed to the published version of the manuscript.

**Funding:** This research received no external funding.

**Data Availability Statement:** All data related to this study are publicly available.

**Acknowledgments:** We appreciate the help and the comments from the reviewers and the editors.

**Conflicts of Interest:** The authors declare no conflict of interest.

## Abbreviations

The following abbreviations are used in this manuscript:

| | |
|---|---|
| UBT | Upside-down bathtub-shaped |
| GEC | Generalized exponentiated composite distributions |
| pdf | Probability density function |
| CDF | Cumulative distribution function |
| EP | Exponential-Pareto distribution |
| EEP | Exponentiated exponential-Pareto distribution |
| LRT | Likelihood ratio test |
| MLE | Maximum likelihood estimates |
| AIC | Akaike information criterion |
| BIC | Bayesian information criterion |
| AICc | Corrected Akaike information criterion |
| cAIC | Consistent Akaike information criterion |
| KS | Kolmogorov–Smirnoff |

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
