# Peer review of "A New Insight into Reliability Data Modeling with an Exponentiated Composite Exponential-Pareto Model"

_applsci, doi:10.3390/app13010645_

Round 1

Reviewer 1 Report

Please consider the attached pdf

Author Response

Dear reviewer:

Please see the word file for the responses. Thank you so much!

Best,

Malwane M.A. Ananda

Reviewer 2 Report

After reading the paper carefully, I recommend rejection for the following reasons:

1. The novelty of the paper is weak. 

2. The motivations of this this paper not clear. 

3. The idea of the paper is very weak and poor.

4. The statistical properties Section is very poor. Moreover, more details should be listed. The authors discussed only moments, HRF, and quantile function for some special models. 

5. Simulation Section is very poor.

6. What is the benefit of discussing the error rates for likelihood ratio test and asymptotic Wald’s test in real data analysis?. 

7. The authors wrote the laws of the hazard function and other concepts several times along the paper, why?.

8.  The authors listed all figures without any details about them. 

9. In Figure 5, the authors used a mix of small and capital letters: "Figure 5. Power Comparison of the Likelihood Ratio Test and Asymptotic Wald’s Test (Red line represents the asymptotic Wald’s test and the black line represents the LRT.

10. The captions for Tables 2 and 3, the authors used a mix of small and capital letters.

11. The authors did not discuss the behavior of data based on non-parametric plots. 

12. In application Section, why the authors did not use the extension of Pareto model to discuss the fitness property?.  The authors used other models "IR and TIR" and only used one extension "weak extension".

13. The authors did discuss the parametric "empirical" plots to prove the fitness such as Probability-Probability and histogram.

14. The authors did not derive the identifiability property of the model.

16. For reliability properties, the authors did not discuss any property except one property "hazard rate", and the study not complete. Statistical literature has many reliability characteristics such as mean past life, mean residual life, availability, maintenance, mean time to failure, among others.   

Author Response

Dear reviewer:

Please see the word file for the responses. 

Best,

Malwane M.A. Ananda

Reviewer 3 Report

REVIEW

Title of the paper: A New Insight into Reliability Data Modeling with an Exponentiated Composite Exponential-Pareto Model

Manuscript Number: applsci-2063022

General conclusion: Major Revision.

Comments

After carefully reading the proposed paper, this paper contains an interesting proposal; my overall impression is that the manuscript presents some results that could be useful in practice. I have a good opinion about this work and recommend its acceptance after addressing the following aspects:

My comments are:

1.   What is the type of the parameters c and  given in equation 1.

2.   The function  should be replaced by  and so in all paper. The symbol “ | ” should be replaced by “ ; ”.

3.      The authors should put “.” in the end of the equations 3, 5, 6, 7, 8, … . Please revise the end of all equations in the paper.

4.      More information around Figure 2 should be reported.

5.      In simulation study, I note that the value of LRT and Wald’s Test sometimes increases and sometimes decreases when the sample size increases, are this correct?

6.      In Figure 7., what is the mining of the “95% CI of Empirical Survival Curve”. How you can compute it from the data set

7.      In Table 2 and 3, the Kolmogorov-Smirnov (KS) statistic and its p-value should be used to compare between the tested models.

8.      Some important figures should be added such that the P-P plots for data set, the estimated PDFs with histogram, box plot for the data set and TTT plot.

9.        At least another data should be added in the paper.

Author Response

Dear Reviewer:

Please see the word file for the responses. Thank you so much!

Best,

Malwane M.A. Ananda

Round 2

Reviewer 2 Report

All comments have been done.

Reviewer 3 Report

Thanks a lot for your response to my comments.